# Electrical Properties and Lifetime Prediction of an Epoxy Composite Insulation Material after Hygrothermal Aging

**DOI:** 10.3390/polym15081942

**Published:** 2023-04-19

**Authors:** Yan Yang, Jielin Ma, Malvern Yap, Qi Wang, Wen Kwang Chern, Yi Shyh Eddy Foo, Zhong Chen

**Affiliations:** 1SP Group–NTU Joint Laboratory, School of Electrical and Electronic Engineering, Nanyang Technological University, Singapore 639798, Singapore; yan.yang@ntu.edu.sg (Y.Y.); eddyfoo@ntu.edu.sg (Y.S.E.F.); 2Singapore Power Group, Singapore 349277, Singapore; 3School of Materials Science and Engineering, Nanyang Technological University, Singapore 639798, Singapore

**Keywords:** aging, humidity, lifetime prediction, dielectric loss, breakdown strength

## Abstract

In this study, we conducted the hygrothermal aging of an epoxy composite insulation material at 95% relative humidity (RH) and temperatures of 95 °C, 85 °C, and 75 °C. We measured electrical properties, including volume resistivity, electrical permittivity, dielectric loss, and breakdown strength. It was found to be impossible to estimate a lifetime based on the IEC 60216 standard, because it uses breakdown strength as its criterion even though breakdown strength hardly changes in response to hygrothermal aging. In analyzing variations in dielectric loss with aging time, we found that significant increases in dielectric loss correlated well with lifetime prediction based on the mechanical strength of the material, as described in the IEC 60216 standard. Accordingly, we propose an alternative lifetime prediction criterion by which a material is deemed to reach its end of life when its dielectric loss reaches 3 and 6–8 times the unaged value at 50 Hz and low frequencies, respectively.

## 1. Introduction

Epoxy resin is the most common insulating material in distribution switchgear. The operating conditions of epoxy resin-based components, including insulators, current and voltage transformers, and bushings, are significantly influenced by the insulating conditions. During operation, the aging of epoxy resin is subjected to multiple stresses, including electrical, thermal, and mechanical stresses [1,2]. For medium-voltage systems in tropical regions, the aging of epoxy resin under combined high temperatures and high humidity levels should be seriously considered.

The thermal endurance properties of electrically insulating materials are well-described in the IEC 60216 series [3,4]. Based on physicochemical models, the Arrhenius equation can be used to evaluate the aging rate of insulating materials after thermal aging. For aging under combined high temperatures and high humidity levels, namely, hygrothermal aging, the effects of water, including the processes of water absorption, decomposition, and resin hydrolysis, should be considered [5]. Thus, a lifetime prediction based on the electrical properties of an epoxy after hygrothermal aging might be different from that after thermal aging.

The influence of hygrothermal aging or water absorption on the electrical properties of epoxy resin has attracted much attention [6,7]. Hygrothermal effects on epoxy resin have been attributed to water molecules attached to epoxy resins through hydrogen bonding [8]. Two types of bound water are found in epoxy resins, depending on the bond complex and activation energy. The first corresponds to a water molecule that forms a single hydrogen bond with the epoxy resin network; the other refers to a water molecule that forms multiple hydrogen bonds with the resin network. In past studies, hygrothermal aging was normally achieved by immersing an epoxy in distilled water at different temperatures, which is quite different from the actual operating conditions of electrical equipment. In the case of medium-voltage electrical systems in tropical regions, hygrothermal aging is more representative of actual environmental conditions. In addition to water absorption, processes such as decomposition and resin hydrolysis should also be investigated.

For lifetime predictions according to the IEC 60216 series, a reduction in electrical breakdown strength to half its initial value is usually used as the criterion for determining the lifetime endpoint for each aging condition. However, under hygrothermal aging, the aging temperature is less than 100 °C, which is much lower than the glass transition temperature, and the breakdown strength might not drop that much because of the presence of a large amount (typically > 50% wt.%) of filler particles in the epoxy composite. Thus, there is a need to find alternative indicators to predict the lifetime of hygrothermal-aged epoxy composite insulator materials.

In this research, we chose a commercial epoxy formula from Huntsman that contains mildly abrasive and mechanically reinforcing fillers. This formula is recommended for the manufacture of electrically insulating components. The electrical properties of the epoxy resin after hygrothermal aging were studied.

## 2. Materials and Methods

A two-part Huntsman epoxy resin composite was chosen in this study. Part A was ARALDITE^®^ CW 229 CI Resin (Huntsman, The Woodlands, TX, U.S.), which contains 50–70 wt.% wollastonite (Ca(SiO_3_)) and 30–50 wt.% diglycidyl ether of bisphenol A (DGEBA). Part B was ARADUR^®^ HW 229 CI Hardener (Huntsman, The Woodlands, TX, U.S.), which contains 20–30 wt.% anhydride and 10–20 wt.% modified anhydride. The two parts were mixed via contentiously magnetic stirring in a 1:1 mass ratio in an oil bath of 65 °C under vacuum until the mixture was homogeneous and free of air bubbles. Then, the degassed mixture was immediately poured into preheated disk molds and cured at 100 °C for 4 h, followed by post-curing at 135 °C for 12 h. The disc samples had a radius of 100 mm and a thickness of 2 mm ± 5%, as shown in Figure 1. Before tests, the samples were conditioned in a dry container at 25 °C and 50% RH for 16 h. More details can be found in [9,10].

The hygrothermal aging was conducted inside a climatic temperature chamber Memmert (Schwabach, Germany) CTC256 set at 95% relative humidity (RH) and temperatures of 95 °C, 85 °C, and 75 °C. Detailed aging conditions are shown in Table 1.

The electrical properties of the unaged and aged samples, including volume resistivity, electrical permittivity, dielectric loss, and AC breakdown strength, were measured under various aging conditions.

### 2.1. Volume Resistivity

The volume resistivity was measured by using a Keithley (Solon, OH, USA) 6517B electrometer, which was equipped with the 8009 test fixture, according to IEC 62631-3-1 [11]. The guarded electrode structure was used to avoid the leakage current. Images of the equipment, connection, and electrode structure are shown in Figure 2. The final values were obtained by averaging the test values of ten samples. The applied voltage was 500 V, and the test duration was 60 s.

### 2.2. Electrical Permittivity and Dielectric Loss

The electrical permittivity and dielectric loss were measured via frequency domain dielectric spectra, as shown in Figure 3. Measurements were obtained at room temperature in the frequency range from 10^−3^ to 10^3^ Hz by using the MeggerIDAX 300 (Dover, UK) insulation diagnostic system equipped with the Keithley 800 test fixture. The applied voltage was 140 Vrms.

### 2.3. Breakdown Strength

The breakdown strength was measured by using the Hipotronics D149-DI AC dielectric breakdown tester (Shelton, CT, USA), according to IEC 60243-1; see Figure 4a. Each sample was immersed in silicone oil (DOW XIAMETERTM PMX-561, Midland, MI, USA) during the test to avoid surface flashover. The electrodes consisted of two copper cylinders with round edges and a radius of 3 ± 0.2 mm. The upper electrode was 25 ± 1 mm in diameter and 25 mm in height. The lower electrode was 75 ± 1 mm in diameter and 15 mm in height, as shown in Figure 4b. The AC breakdown voltages of ten samples were tested, and the values of breakdown strength were estimated with the Weibull distribution according to IEC 62539 [12]. The ramping rate of the test voltage was set to 100 V/s.

## 3. Results

### 3.1. Volume Resistivity

The volume resistivity values of samples under different aging conditions are shown in Figure 5. The initial value of volume resistivity for the unaged sample was in the order of magnitude of fifteen Ω·cm. After hygrothermal aging at 95% relative humidity (RH) and a temperature of 95 °C, the volume resistivity decreased with aging time, dropping one order of magnitude to fourteen, as shown in Figure 5a. The trends were similar for samples aged under 95% relative humidity (RH) at 85 °C and 75 °C, as shown in Figure 5b and 5c, respectively, while the decreasing rate was slower at lower aging temperatures. As found in our previous study [9], the drop in volume resistivity is attributed to moisture absorption during hydrothermal aging. In addition, O–H groups are generated by the hydrolysis of the resin during hygrothermal aging, which makes ionic transportation easier with lower barriers, thus resulting in lower volume resistivity values [13].

### 3.2. Electrical Permittivity and Dielectric Loss

The frequency-domain dielectric spectra are shown in Figure 6 and Figure 7. After hygrothermal aging, the dielectric loss increased in the low-frequency range (10^−3^–1 Hz). Under the 95 °C and 95% RH aging conditions, the dielectric loss significantly increased in the low-frequency range after hygrothermal aging for 1000 h. For the same aging time, the values of electrical permittivity and dielectric loss were lower at lower aging temperatures. However, for samples subjected to thermal aging at 95 °C without moisture control, changes in dielectric loss were negligible even after being aged for 1006 h, as shown in Figure 8, compared with samples aged at 95 °C and 95% RH. This implies the importance of water uptake in the dielectric loss of an insulation material. In addition, the dielectric loss in the low-frequency range is largely dependent on the polarization of charge carrier transport and hopping in the bulk when the measured temperature is lower than the glass transition temperature [14,15,16], as tested in this study. Our results were consistent with those of volume resistivity studies where easier ionic transportation results in higher dielectric losses with increasing aging time and aging temperatures.

### 3.3. AC Breakdown Strength

The Weibull distribution is the most commonly used tool for describing the failure of solid insulation caused by the weakest link breakdown. The two-parameter Weibull distribution was used in this work.
(1)$$F\left(E\right)=1-\exp\left\{-{\left(\frac{E}{\alpha }\right)}^{\beta }\right\}$$
where *E* is the measured breakdown strength, *F*(*E*) is the probability of failure at an electrical field less than or equal to *E*, *α* is the scale parameter, and *β* is the shape parameter. The scale parameter *α* represents the electrical field at which the failure probability is 0.632 (IEC 62539 [12]). The shape parameter *β* is a measure of the range of the failure voltages.

According to Equation (1), the curve ln(−ln(1 − *F*)) vs. ln(*E*) should be linear, which is called the Weibull plot. The slope of the Weibull plot is equal to the shape parameter *β*. The value of the scale parameter *α* can be calculated based on the fitting linear equation of the Weibull plot by setting *F* equal to 0.632. The Weibull plots of samples under various aging conditions are shown in Figure 9.

The Weibull parameters are shown in Table 2. Breakdown strength can be estimated with parameter *α*. It should be noted that there was no obvious change in the AC breakdown strength against aging time based on experimental results. The breakdown strength of each group was around 16 kV/mm.

## 4. Discussion

### 4.1. Lifetime Prediction

According to IEC 60216-2 [4], the lifetime estimation of electrically insulating materials after thermal aging can be based on both mechanical and electrical strength properties. The recommended properties for epoxy resin include flexural strength, tensile strength, and breakdown voltage. When these values reach the threshold defined by the criterion, the corresponding aging time can be considered the lifetime under certain aging conditions. Usually, a lifetime can be obtained when the breakdown strength drops to half its initial value, namely, the value of an unaged sample. However, as shown in Table 2, breakdown strength does not drop with aging time. Thus, it seems impossible to estimate a lifetime based on the criterion set in the IEC 60216. From the mechanical perspective, a lifetime can be estimated by taking mechanical strength as the criterion [10]. However, from the electrical point of view, we needed to analyze if there are any parameters other than breakdown strength that can be considered aging indicators. Next, variations in dielectric parameters with aging time were studied.

Once the lifetime was obtained at each aging condition, namely, 95% relative humidity (RH) and temperatures of 75 °C, 85 °C, and 95 °C, a straight line could be obtained via the Arrhenius equation, and then the lifetime at each operating temperature could be predicted based on the linear equation.

### 4.2. Variations in Dielectric Parameters with Aging Time

From the dielectric spectra shown in Figure 6 and Figure 7, we can choose the values at certain frequencies to study variations in dielectric parameters with aging time. The variations in normalized electrical permittivity and dielectric loss (compared with the unaged sample) are shown in Figure 10 and Figure 11, respectively. The chosen frequencies were 0.001, 0.01, 0.1, 1, and 50 Hz. Both the electrical permittivity and dielectric loss increased with aging time, and increases were more significant at lower frequencies. However, when the aging temperature was 75 °C, which was not high enough to cause significant aging, the increase in electrical permittivity was not stable, as shown in Figure 10c. However, the rise in dielectric loss was more significant and stable, as shown in Figure 11.

Unlike electrical or mechanical breakdown strength, these dielectric parameters increased with aging time. Thus, it is impossible to find a half-value point to estimate a lifetime, as recommended in IEC 60216-2 [4]. To study the law of variation, we applied least squares fitting to the tan *δ–*aging time curves by using the second-order polynomial function. The *R*-squared values of fitting were generally greater than 0.85, which indicated good fitting for the samples aged under 95% relative humidity (RH) and 95 °C, 85 °C, and 75 °C, as shown in Figure 12, Figure 13 and Figure 14, respectively.

As shown in Figure 12, a sudden rise of the curve appears at around 1000 h of aging time, indicating the lifetime endpoint, which is consistent with results obtained with mechanical analysis [10]. In addition, at the lifetime endpoint, the value of tan *δ* was 3 and 6 times the unaged value at 50 Hz and low frequencies, respectively.

As shown in Figure 13, there is also a rise from 1500 to 2000 h on the tan *δ*–aging time curve, but it is not as obvious as that under aging at 95 °C. The 1500–2000 h period was also consistent with the estimated lifetime obtained with mechanical analysis [10]. In addition, at the lifetime endpoint, the value of tan *δ* was 3 and 6–8 times the unaged value at 50 Hz and low frequencies, respectively.

As shown in Figure 14, no obvious rise can be observed because the lifetime estimated with mechanical analysis was about 3700 h [10] and we conducted a relatively shorter aging period. After being aged for 2500 h, the value of tan *δ* was 2.5 and 5–6 times the value of the unaged sample at 50 Hz and low frequencies, respectively. Comparing the results for different aging temperatures reveals that if the value of dielectric loss reaches 3 times and 6–8 times the unaged value at 50 Hz and low frequencies, respectively, the material has approached its lifetime endpoint.

## 5. Conclusions

In this study, commercially available epoxy resin composite samples were aged at 95% relative humidity (RH) and temperatures of 95 °C, 85 °C, and 75 °C. The electrical properties of samples aged under various aging conditions were studied. It was found that the volume resistivity decreased by an order of magnitude after hygrothermal aging. Electrical permittivity and dielectric loss increased with aging time, especially in the low-frequency range. However, the breakdown strength hardly changed with aging because of the relatively low aging temperatures and high filler content. Thus, it was impossible to estimate lifetimes by using the criterion recommended by the IEC 60216 standard series. Based on the results regarding dielectric loss variation against aging time, it is recommended that a sudden rise in the curve of dielectric loss against aging time can be an indicator of a lifetime endpoint, especially when the value of dielectric loss reaches 3 and 6–8 times the unaged value at 50 Hz and low frequencies, respectively.

## Figures and Tables

**Figure 1 polymers-15-01942-f001:**
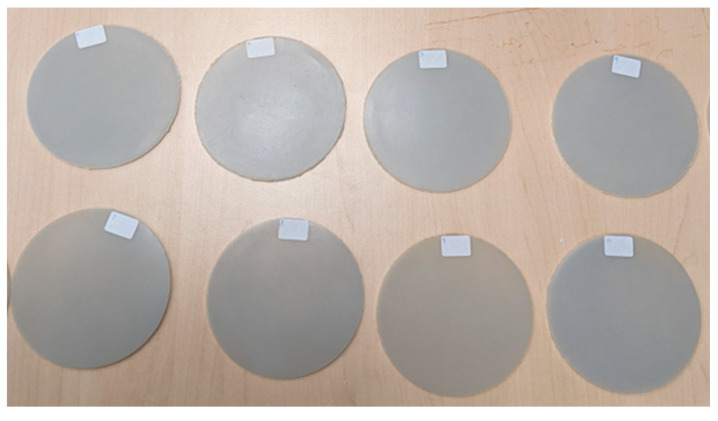
Image of epoxy resin composite samples.

**Figure 2 polymers-15-01942-f002:**
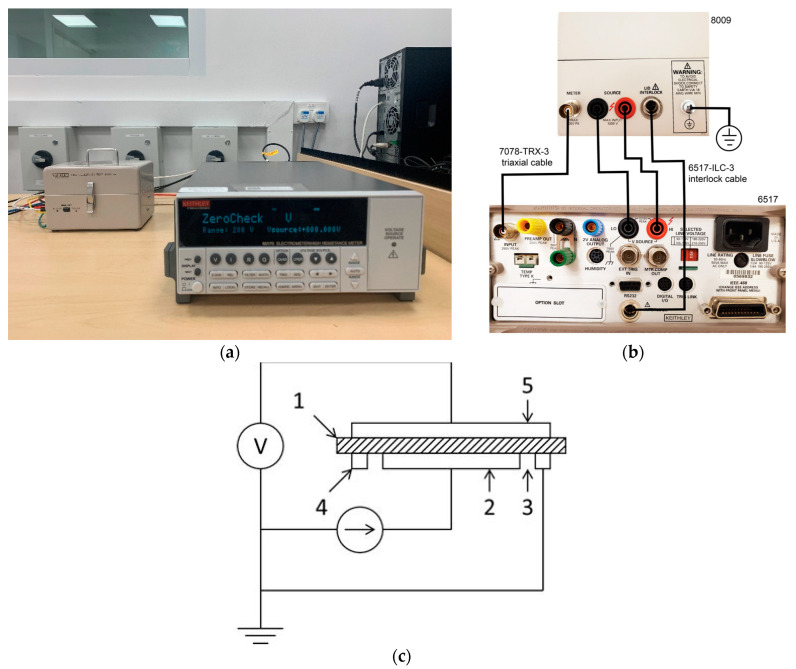
Volume resistivity measurement setup: (**a**) Image of Keithley 6517B and 8009 fixture; (**b**) connection of Keithley 6517B and 8009 fixture; (**c**) structure of guarded electrodes, where 1 is the measuring area, 2 is the lower electrode, 3 is the sample, 4 is the guard electrode, and 5 is the upper electrode.

**Figure 3 polymers-15-01942-f003:**
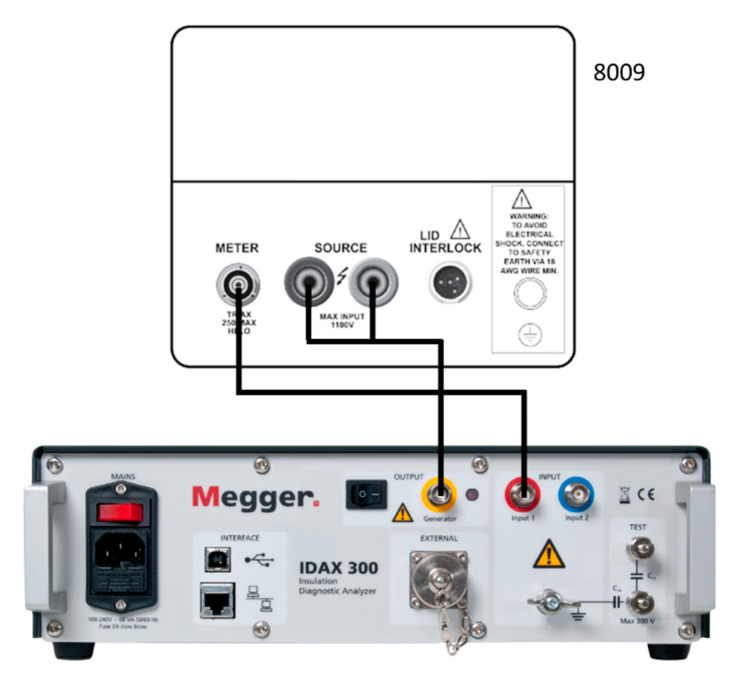
Equipment and connection of frequency domain dielectric spectra measurement setup.

**Figure 4 polymers-15-01942-f004:**
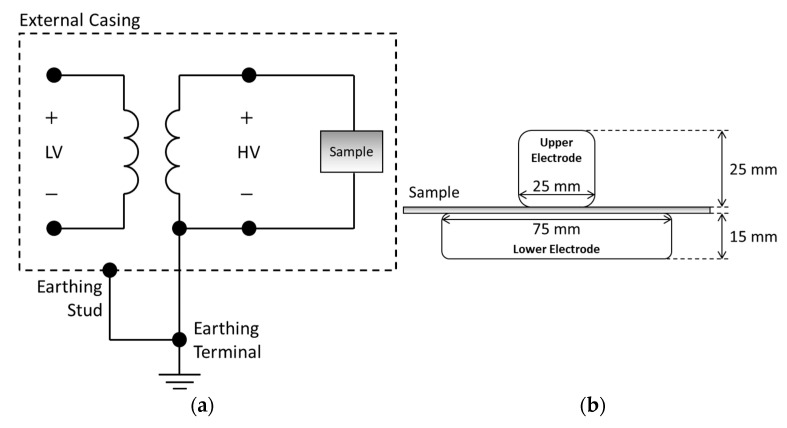
Electrical breakdown strength measurement setup: (**a**) Test circuit; (**b**) electrode structure.

**Figure 5 polymers-15-01942-f005:**
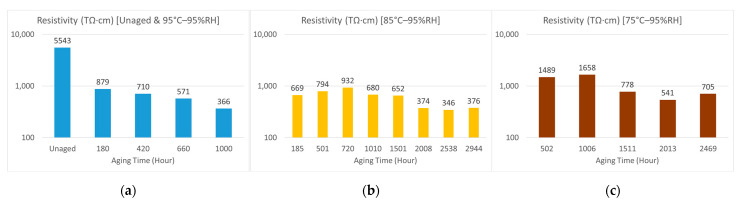
Volume resistivity: (**a**) Unaged samples and samples aged at 95% relative humidity (RH) and a temperature of 95 °C; (**b**) samples aged at 95% relative humidity (RH) and a temperature of 85 °C; (**c**) samples aged at 95% relative humidity (RH) and a temperature of 75 °C.

**Figure 6 polymers-15-01942-f006:**
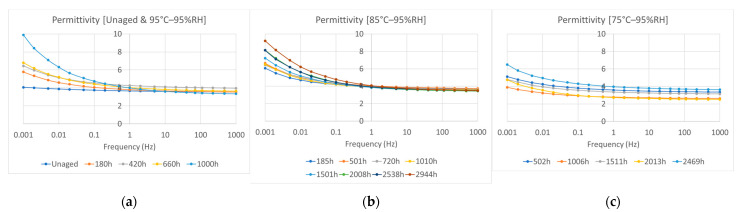
Electrical permittivity: (**a**) Unaged samples and samples aged at 95% relative humidity (RH) and a temperature of 95 °C; (**b**) samples aged at 95% relative humidity (RH) and a temperature of 85 °C; (**c**) samples aged at 95% relative humidity (RH) and a temperature of 75 °C.

**Figure 7 polymers-15-01942-f007:**
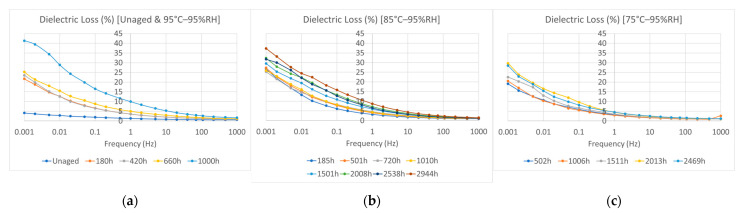
Dielectric loss: (**a**) Unaged samples and samples aged at 95% relative humidity (RH) and a temperature of 95 °C; (**b**) samples aged at 95% relative humidity (RH) and a temperature of 85 °C; (**c**) samples aged at 95% relative humidity (RH) and a temperature of 75 °C.

**Figure 8 polymers-15-01942-f008:**
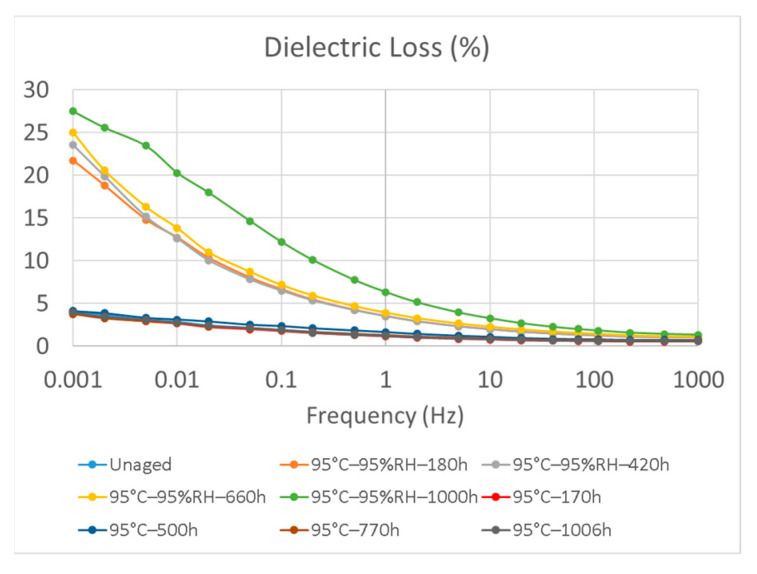
The dielectric loss of samples aged at 95 °C with and without moisture control.

**Figure 9 polymers-15-01942-f009:**
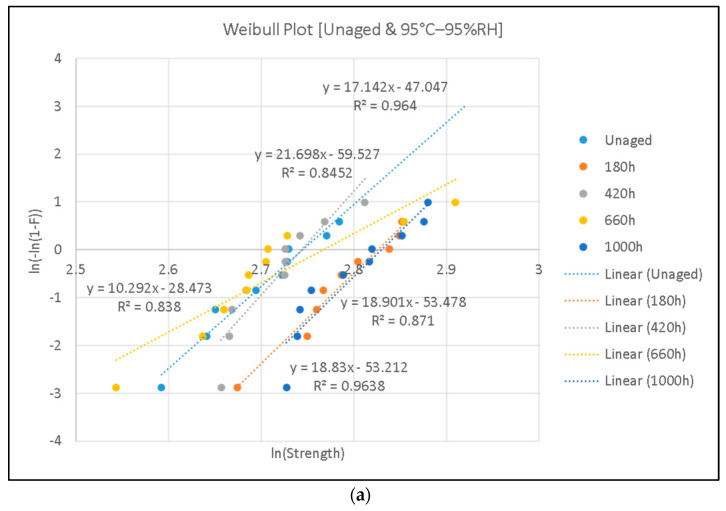
Weibull plots of breakdown strength measurements: (**a**) Unaged samples and samples aged at 95% relative humidity (RH) and a temperature of 95 °C; (**b**) samples aged at 95% relative humidity (RH) and a temperature of 85 °C; (**c**) samples aged at 95% relative humidity (RH) and a temperature of 75 °C.

**Figure 10 polymers-15-01942-f010:**
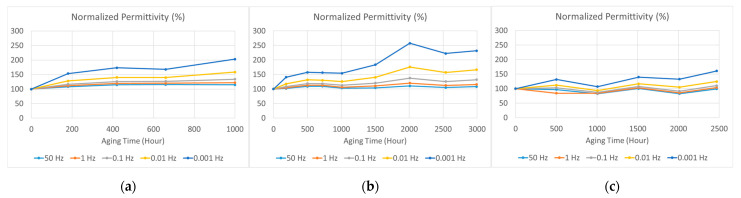
The variation in normalized electrical permittivity for samples aged at various aging conditions and frequencies of 0.001, 0.01, 0.1, 1, and 50 Hz: (**a**) Unaged samples and samples aged at 95% relative humidity (RH) and a temperature of 95 °C; (**b**) samples aged at 95% relative humidity (RH) and a temperature of 85 °C; (**c**) samples aged at 95% relative humidity (RH) and a temperature of 75 °C.

**Figure 11 polymers-15-01942-f011:**
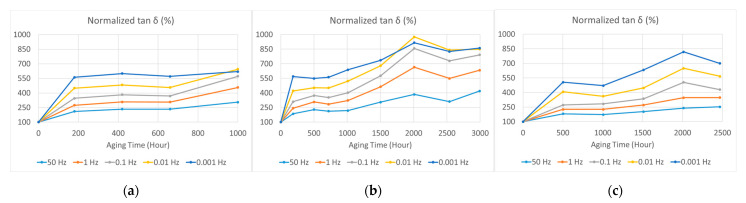
The variation in normalized dielectric loss for samples aged at various aging conditions and frequencies of 0.001, 0.01, 0.1, 1, and 50 Hz: (**a**) Unaged samples and samples aged at 95% relative humidity (RH) and a temperature of 95 °C; (**b**) samples aged at 95% relative humidity (RH) and a temperature of 85 °C; (**c**) samples aged at 95% relative humidity (RH) and a temperature of 75 °C.

**Figure 12 polymers-15-01942-f012:**
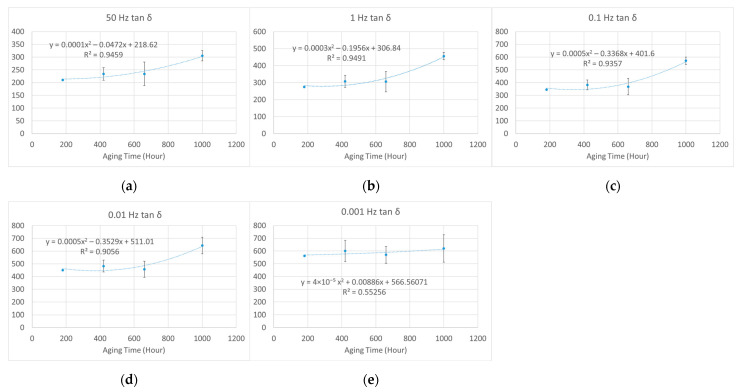
The second-order polynomial fitting of normalized dielectric loss (%) against aging time at different frequencies (aged at 95% relative humidity (RH) and 95 °C): (**a**) 50 Hz; (**b**) 1 Hz; (**c**) 0.1 Hz; (**d**) 0.01 Hz; (**e**) 0.001 Hz.

**Figure 13 polymers-15-01942-f013:**
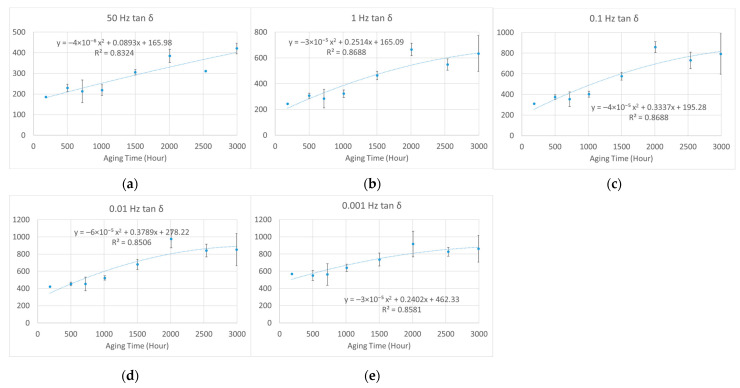
The second-order polynomial fitting of normalized dielectric loss (%) against aging time at different frequencies (aged at 95% relative humidity (RH) and 85 °C): (**a**) 50 Hz; (**b**) 1 Hz; (**c**) 0.1 Hz; (**d**) 0.01 Hz; (**e**) 0.001 Hz.

**Figure 14 polymers-15-01942-f014:**
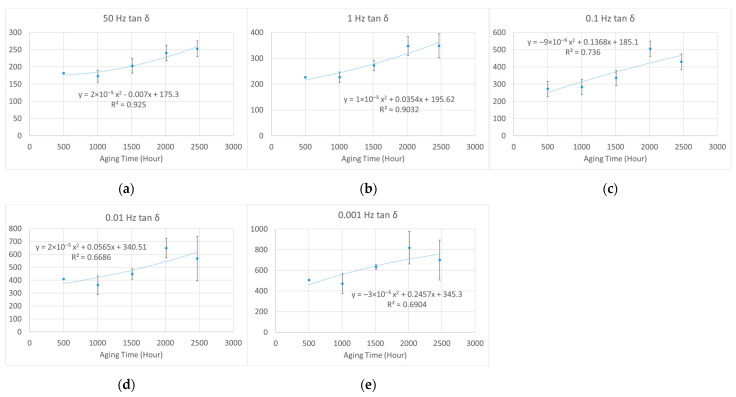
The second-order polynomial fitting of normalized dielectric loss (%) against aging time at different frequencies (aged at 95% relative humidity (RH) and 75 °C): (**a**) 50 Hz; (**b**) 1 Hz; (**c**) 0.1 Hz; (**d**) 0.01 Hz; (**e**) 0.001 Hz.

**Table 1 polymers-15-01942-t001:** Aging conditions.

Aging Temperature (°C)	Aging Time (Hours)
95	180, 420, 660, 1000
85	185, 501, 720, 1010, 1501, 2008, 2538, 2944
75	502, 1006, 1511, 2013, 2469

**Table 2 polymers-15-01942-t002:** Weibull parameters of samples under different aging conditions.

Aging Conditions	Scale *α* (kV/mm)	Shape *β*
Unaged	15.56	17.14
95 °C–95%RH–180 h	16.88	18.83
95 °C–95%RH–420 h	15.54	21.70
95 °C–95%RH–660 h	15.90	10.29
95 °C–95%RH–1000 h	16.93	18.90
85 °C–95%RH–185 h	15.08	10.59
85 °C–95%RH–501 h	15.64	8.34
85 °C–95%RH–2008 h	16.75	7.46
85 °C–95%RH–2944 h	15.28	9.15
75 °C–95%RH–1006 h	17.02	10.00
75 °C–95%RH–1511 h	16.57	9.22
75 °C–95%RH–2013 h	16.73	9.34
75 °C–95%RH–2469 h	16.24	14.28

## Data Availability

Date are available from the corresponding author upon reasonable request.

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
