# Peer review of "Electrical Properties and Lifetime Prediction of an Epoxy Composite Insulation Material after Hygrothermal Aging"

_polymers, 2023, doi:10.3390/polym15081942_

Round 1
Reviewer 1 Report
This paper investigated the electrical properties of an epoxy material after a series of hydrothermal aging conditions, and proposed a criterion to determine the lifetime. Generally the design is appropriate, the results are reasonable, it can be accepted after making some improvements:
1. It is recommended to provide a detailed description of the tested epoxy composites? For example, what are the exact components.
2. The details of the test specimen should also be provided, how it is made, an image, etc.
3. The author should provide more information about the test methods. Such as, how the sample is connected to the test machine, the electrodes, the electrode location, etc. Images are highly recommended.
4. Line 66, what is the specific model of the climatic temperature chamber.
5. Throughout the paper, the font in the figures should be enlarged. It is very hard to read in the print version of the manuscript.
6. The conclusion part should be re-wrote to make it more integrated and logical, it is recommended to write in one paragraph rather than several bullet points.
The overall quality of English Language is fine. Just provide more details of the sample preparation, test methods, and conclusion.
Author Response
Thank you for your comments and suggestions. Please find our detailed response in the uploaded file.

Reviewer 2 Report
The paper presents the study about electrical properties and lifetime prediction of epoxy composite insulation material after hygrothermal aging. Authors made investigations about hygrothermal aging of an epoxy composite insulation material at 95% relative humidity (RH) and the temperature of 95 °C, 85 °C, and 75 °C, respectively. They measured volume resistivity, permittivity, dielectric loss, and breakdown strength. In their opinion, it is impossible to estimate the lifetime based on the criterion IEC 60216 which uses breakdown strength because the strength changes with hygrothermal aging. Authors propose an alternative lifetime prediction criterion by which the material is deemed as reaching the end of life when the dielectric loss reaches 3 times, what is very bold thesis in my mind.
Dear author, thank you very much for interesting paper about aging of epoxy insulation material. I put some comments and questions.
Comments:
1. The introduction is well organized. Authors describe many important properties of epoxy, which play important role in case of electric power devices as high voltage insulation.
2. Chapter 2 – authors describe object of investigation as epoxy composite. What was a composite? Epoxy with resin or something different. Please explain in details the object of research. Also, please tell something about the way of preparation. What I remember, the preparation of epoxy material is very complicated, especially in case of avoid small bubble of airs.
3. Chapter 2.2 - “Permittivity …”. Correct name is “electrical permittivity”.
4. Fig.1. – why did authors use different period of aging for (a), (b) and (c) cases? It is not mistake but it is very difficult to compare the figures.
5. Fig.2. – permittivity as a function of what? I can imagine that … as a function of frequency. Please complete the description of X axis.
6. Fig.3. – see comments for Fig.2. Please complete Fig (b) and (c).
7. Fig.5. – it is difficult to analyze this figures, because the details are too small. I would suggest to cut this figure to 3 new figures. The same comments are for other multi figures.
8. Proposed method to evaluate the state of epoxy as insulation material is interesting and bold. I am not sure that electrical strength is not important to estimate the level of epoxy insulation. Maybe connection two methods would be good solution. I mean traditional method together with method proposed by authors.
Author Response

(The authors gave the same response as above.)
